# Macrolactin XY, a Macrolactin Antibiotic from Marine-Derived *Bacillus subtilis* sp. 18

**DOI:** 10.3390/md22080331

**Published:** 2024-07-23

**Authors:** Yao Xu, Yihao Song, Yaodong Ning, Song Li, Yingxin Qu, Binghua Jiao, Xiaoling Lu

**Affiliations:** Department of Biochemistry and Molecular Biology, College of Basic Medical Sciences, Naval Medical University, Shanghai 200433, China; nmuxy@smmu.edu.cn (Y.X.); fcs032@163.com (Y.S.); ning_yd@sina.com (Y.N.); lisonglisong0716@163.com (S.L.); 1020609684@smmu.edu.cn (Y.Q.)

**Keywords:** macrolactins, antibiotic, antibacterial mechanism, cell membrane integrity

## Abstract

Two new compounds, macrolactin XY (**1**) and (5*R*, 9*S*, 10*S*)-5-(hydroxymethyl)-1,3,7-decatriene-9,10-diol (**2**), together with nine known compounds (**3**–**11**) were isolated from the marine *Bacillus subtilis* sp. 18 by the OSMAC strategy. These compounds were evaluated for antibacterial activity against six tested microorganisms. Compounds **1**–**5** and **7**–**10** showed varied antibacterial activity, with the minimum inhibitory concentration (MIC) ranging from 3 to 12 μg/mL. Macrolactin XY (**1**) was found to possess superior antibacterial activity, especially exhibiting significant effectiveness against *Enterococcus faecalis*. The antibacterial activity mechanism against *E. faecalis* was investigated. The mechanism may disrupt bacterial cell membrane integrity and permeability, and also inhibit the expression of genes associated with bacterial energy metabolism, as established by the experiments concerning cell membrane potential, SDS-PAGE electrophoresis, cell membrane integrity, and key gene expressions. This study offers valuable insights and serves as a theoretical foundation for the future development of macrolactins as antibacterial precursors.

## 1. Introduction

*E. faecalis* is a conditionally pathogenic bacterium that is commonly found in the normal flora of the intestinal tract and frequently implicated in hospital-acquired infections. Its clinical infections typically manifest as periodontitis, urinary tract infections, abdominal and pelvic trauma, and post-surgical infections [1,2,3]. Given its infectious nature and the extensive research conducted on its pathogenic mechanisms, *E. faecalis* serves as a valuable reference for diagnosing and treating various diseases. The current arsenal of medications used in treating *E. faecalis* infections primarily comprises penicillins and aminoglycoside antibiotics. Penicillins, the preferred drug for the treatment of *E. faecalis* infection, exhibit high sensitivity towards the pathogen, albeit with some resistance stemming from the bacterium’s robust cell walls [4,5]. Moreover, prolonged antibiotic usage can perturb the intestinal flora balance, manifesting in gastrointestinal symptoms like anorexia, nausea, and dyspepsia. When penicillins are ineffective, aminoglycosides can be administered in tandem, albeit with the caveat of potential side effects such as ototoxicity and nephrotoxicity [6,7], necessitating cautious use. In light of the current scarcity of targeted therapeutic medications, it is beneficial to explore these therapeutic alternatives.

Macrolactins are natural compounds that feature three distinct diene structures embedded within a 24-membered lactone ring. So far, over 50 macrolactin analogs have been isolated [8]. These compounds exhibit diverse biological activities, including antibacterial, anti-inflammatory, antitumor, and antiviral effects [9,10,11,12], offering a vast array of potential avenues for novel drug development. Antibacterial activity is one of the most superior bioactivities of macrolactins, and they showed a strong effect on *Staphylococcus aureus*, *Bacillus subtilis*, and *Escherichia coli*, which might become new antibacterial lead compounds.

In this study, eleven compounds were obtained from *Bacillus subtilis* sp. 18 (Figure 1), including two new compounds (macrolactin XY (**1**) and (5*R*, 9*S*, 10*S*)-5-(hydroxymethyl)-1,3,7-decatriene-9,10-diol (**2**)) and nine known compounds (**3**–**11**). Subsequently, the antibacterial activity of all compounds was evaluated, and it was found that the new compound, macrolactin XY (**1**), showed significant activity against *E. faecalis*. Given the remarkable antibacterial activity of macrolactins, a preliminary investigation was conducted into its antibacterial mechanism using *E. faecalis* as an indicator bacterium.

## 2. Results and Discussion

### 2.1. Structural Analysis of Compounds

Compound **1** was isolated as a white powder. The molecular formula was determined to be C_25_H_36_O_5_ by HRESIMS, implying eight degrees of unsaturation. The UV spectrum exhibited absorption at *λ*_max_ 228 and 262 nm (Appendix A). The ^1^H-NMR and ^13^C-NMR data are shown in Table 1 (Appendix A). The ^1^H-NMR data displayed two characteristic methyl proton signals at δ_H_ 3.27 (3H, s) and δ_H_ 1.28 (3H, d, *J* = 6.3 Hz). Analysis of the ^13^C-NMR data and DEPT spectrum revealed a total of twenty-five carbons, including two methyls, six methylenes, sixteen methines, and one quaternary carbon signal. Twelve double-bond carbon signals were found at *δ*c 118.2, 143.0, 130.0, 139.6, 135.8, 125.1, 130.0, 128.3, 130.1, 133.3, 129.9, and 135.5. Four carbons connecting oxygen atoms at δ_C_ 69.41, 71.16, 71.41, and 80.37, and one methoxyl carbon signal at δ_C_ 56.39 could be observed from the ^13^C-NMR (Appendix A).

The COSY correlations of H-2/H-3, H-4/H-5, H-6/H-7, H-7/H-8, H-8/H-9, H-12/H-13, H-13/H-14, H-14/H-15, H-15/H-17, H-20/H-21, H-21/H-22, H-22/H-23, H-23/H-24, and H-24/H-25 revealed the presence of five isolated spin systems: C-2/C-3, C-4 and C-6, C-6/C-7/C-8/C-9, C-12/C-13/C-14/C-15 and C-17, and C-20/C-21/C-22/C-23/C-24/C-25 (Figure 2 and Appendix A). The diagnostic HMBC correlations from H-25 to C-24, H-24 to C-1 and C-22/C-23, H-3 to C-1 and C-5, H-9 to C-7 and C-10, H-16 to C-15, H-12 to C-10/C-11 and C-13/C-14, and H-18 to C-15 and C-20 indicated a macrocycle with a 24-membered ring serving as its core structure (Appendix A). Due to the presence of three oxygenated carbon signals, the oxygenated carbons at C-7 and C-13 had to be substituted with a hydroxy group to satisfy the molecular formula, and the carbons at C-15 and C-24 were substituted with methoxyl and methyl, respectively (Figure 2). The ^1^H coupling constant between H-2 and H-3 was 11.6 Hz, while the coupling constant between H-4 and H-5 was 15.2 Hz, indicating that the geometries of C-2 and C-4 were *Z* and *E* configurations. The ^1^H coupling constant between H-8 and H-9 was 15.1 Hz, while the coupling constant between H-10 and H-11 was 9.0 Hz, indicating that the geometries of C-8 and C-10 were *E* and *Z* configurations. The ^1^H coupling constant between H-17 and H-18 was 15.2 Hz, while the coupling constant between H-19 and H-20 was 14.5 Hz, indicating that the geometries of C-17 and C-19 were both *E* configurations. Thus, a planar structure was indicated.

Once the planar structure was established, its relative configuration was addressed by NOESY experiments (Figure 3). The NOESY cross-peaks of H-7/H-10, H-10/H-11, H-11/H-13 and H-10/H-24 indicated the same orientation of these protons. Therefore, the relative structure was determined (Appendix A).

To ascertain the comprehensive absolute configurations of **1**, the quantum chemical electronic circular dichroism (ECD) calculation method was utilized. The negative Cotton effect (CE) at 260 nm and positive cotton effect at 233 nm in the calculated spectrum of the 7*R*, 13*R*, 15*S*, 24*R* enantiomer approximately matched the CE observed in the experimental ECD spectrum of **1** (Figure 4), enabling the assignment of the absolute configuration of **1**, as demonstrated. Finally, compound **1** was named as macrolactin XY. Compared to the other macrolactins, the main difference between macrolactin XY (**1**) and the others was that the latter were hydroxylated at C-15, whereas macrolactin XY (**1**) was methoxylated.

Compound **2** was isolated as a yellow oil. The molecular formula was determined to be C_11_H_18_O_3_ by HRESIMS, implying three degrees of unsaturation. The UV spectrum exhibited absorption at *λ*_max_ 226 nm (Appendix A). The ^1^H-NMR and ^13^C-NMR data are shown in Table 1 (Appendix A). The ^1^H-NMR data displayed one characteristic methyl proton signal at δ_H_ 1.14 (3H, d, *J* = 6.4 Hz); seven olefinic protons signals at δ_H_ 6.35 (1H, dt, *J* = 14.2, 8.5 Hz), 6.15 (1H, dd, *J* = 13.0, 8.5 Hz), 5.67 (1H, m), 5.13 (1H, dd, *J* = 14.3, 1.5 Hz), 5.58 (1H, dd, *J* = 5.5, 13.0 Hz), 5.62 (1H, m), and 4.99 (1H, dd, *J* = 10.1, 1.8 Hz); one oxygen-bearing methylene proton signal at δ_H_ 3.53 (2H, d, *J* = 6.6 Hz); and three aliphatic methine signals, including two methine oxide signals at δ_H_ 3.86 (1H, m), 3.65 (1H, m), and one methine signal at δ_H_ 2.98 (1H, t, *J* = 6.9 Hz). Analysis of the ^13^C-NMR data and DEPT spectrum revealed a total of eleven carbons, including six double-bond carbon signals at *δ*c 116.2, 138.5, 133.6, 135.1, 133.5, and 132.5, three methine carbon signals, one methylene carbon signal, and one methyl carbon signal. Its structural characteristics mean that it is a precursor to terpene synthetic substances, suggesting the potential of producing some terpenes in this strain (Appendix A).

The COSY correlations of H-1/H-2, H-2/H-3, H-3/H-4, H-4/H-5, H-5/H-6, H-5/H-7, H-7/H-8, H-8/H-9, H-9/H-10, and H-10/H-11 revealed the presence of two isolated spin systems: C-1/C-2/C-3/C-4/C-5/C-6 and C-5/C-7/C-8/C-9/C-10/C-11 (Figure 2 and Appendix A). The diagnostic HMBC correlations from H-1 to C-2/C-3, H-4 to C-2/C-3 and C-5/C-6, H-4 to C-7, H-8 to C-9/C-10, and H-11 to C-9/C-10 indicated the presence of a terminal double bond conjugated with a trans double bond, confirming that an oxygen-bearing methylene is linked to this conjugated double bond via a methine group (Appendix A). Due to the presence of three oxygenated carbon signals, C-6 and C-9/C-10 had to be substituted with a hydroxyl group to satisfy the molecular formula. The ^1^H coupling constant between H-3 and H-4 was 13.0 Hz, while the coupling constant between H-7 and H-8 was 13.0 Hz, indicating that the geometries of C-3 and C-7 were both *E* configurations. Thus, the planar structure was determined (Figure 2).

Once the planar structure was established, its relative configuration was addressed by NOESY experiments (Figure 3). The NOESY cross-peaks of H-9/H-10 and H-5/H-10 indicated the same orientation of these protons and that they were located on the same side of the molecule (Appendix A). Therefore, the relative structure was determined. To ascertain the comprehensive absolute configuration of **2**, the quantum chemical electronic circular dichroism (ECD) calculation method was utilized. The negative Cotton effect (CE) at 224 nm and positive CE at 204 nm in the calculated spectrum of the 5*R*, 9*S*, 10*S* enantiomer approximately matched the CE observed in the experimental ECD spectrum of **2** (Figure 4), enabling the assignment of the absolute configuration of **2**, as demonstrated. ECD analysis was also performed on six other isomers of compound **2** (Appendix A), however, the results indicated that none of them matched the experimental ECD spec-trum. Finally, compound **2** was named as (5*R*, 9*S*, 10*S*)-5-(hydroxymethyl)-1,3,7-decatriene-9,10-diol.

In addition, nine known compounds (**3**–**11**) were also isolated as metabolites of *Bacillus subtilis* sp. 18 (Figure 1). These compounds were identified through a comparison of their spectral data with the spectroscopic data reported in the relevant literature, and all of them were classified as follows: macrolactin F (**3**), macrolactin A (**4**), 3-phenyl-2,4-pyrrolidinedione (**5**), cyclo (Phe-Thr) (**6**), tryptophol (**7**), 2,3,3-trimethylindolenine (**8**), 3-hydroxy-3-methyl-2(3H)-benzofuranone (**9**), l-leucyl-l-proline lactam (**10**), and cyclo (Pro-Val) (**11**).

### 2.2. Antibacterial Activity

#### 2.2.1. Determination of MIC and MBC of Macrolactin XY against Test Microorganisms

Compounds **1**–**11** exhibited inhibitory activity against a range of microorganisms, including *S. aureus*, *B. subtilis*, *E. coli*, *E. faecalis*, *Vibrio traumaticus*, and *Vibrio parahaemolyticus*. As shown in Table 2, compounds **1**–**5** and **7**–**10** showed varied antibacterial activity, with MICs ranging from 3 to 12 μg/mL. Among them, macrolactin XY (**1**), macrolactin F (**3**), and macrolactin A (**4**) showed strong antibacterial effects, especially macrolactin XY (**1**), demonstrating a significant inhibitory effect (MIC 3 μg/mL and MBC 12 μg/mL) on *E. faecalis*.

Among the three isolated macrolactins, macrolactin XY (**1**) demonstrated superior activity compared to macrolactins A and F, particularly against *E. faecalis*. The analysis of known structure–activity relationships revealed that macrolactins exhibit great antibacterial activity when the C-15 position is linked to a hydroxyl group [8,13,14]. This suggests that the methoxyl group at the C-15 position contributes significantly to antibacterial effects, identifying the C-15 site as a crucial active site for the antibacterial activity of macrolactins.

#### 2.2.2. Analysis of Bacterial Growth Curve

To further validate the antibacterial effect of macrolactin XY (**1**) against *E. faecalis*, a detailed investigation on the impact of macrolactin XY (**1**) on the specific growth rate of *E. faecalis* was conducted, as presented in Table 3. Additionally, a growth curve was plotted to illustrate the interaction between macrolactin XY (**1**) and *E. faecalis*, as depicted in Figure 5.

Upon examination of the growth curves, it became evident that the OD values of the control group exhibited a rapid increase, particularly in comparison to the treated group, and stabilized after approximately 20 h. This observation was primarily attributed to the bacteria’s growth cycle. Notably, when the concentration of the sample solution was set at the MIC, bacterial growth was delayed over time. The resulting OD value was significantly lower than that of the control group (*p* < 0.05). Furthermore, the specific growth rate of *E. faecalis* decreased significantly (*p* < 0.05) with the effect of macrolactin XY (**1**), indicating its inhibitory effect on bacterial growth.

### 2.3. Antibacterial Mechanism

#### 2.3.1. Effect on Bacterial Cell Membrane Potential

As depicted in Figure 6a, alterations in the membrane potential of *E. faecalis* exerted a profound influence on the metabolic activity of the bacteria [15]. Specifically, a decrease in fluorescence intensity served as an indicator of a reduced potential difference between the intracellular and extracellular environments [16]. Notably, the mean fluorescence intensity decreased by 23.9% when the concentration of macrolactin XY (**1**) approached the MIC level, compared to both the control and negative control groups. This decrement was statistically significant (*p* < 0.0001). In contrast, there was no significant difference between the control and negative control groups (*p* > 0.05), indicating that macrolactin XY (**1**) specifically targets bacterial membrane potential, ultimately affecting metabolic activity.

#### 2.3.2. Results of SDS-PAGE Electrophoresis of Bacterial Proteins

Proteins serve as the fundamental building blocks of all life, and are essential for maintaining normal metabolic processes and facilitating the transportation of diverse substances throughout the body. The expression level of proteins provides a valuable indicator of the state of cellular life [17]. Bacteria harbour numerous crucial macromolecular proteins, such as glucose transporters, a class of transmembrane protein situated on the bacterial cell membrane. These are vital regulators of glucose’s transmembrane transport, playing a pivotal role in bacteria’s energy acquisition and metabolic processes [18]. Other enzymes that are integral to the biosynthesis of peptidoglycan in the bacterial cell wall, like penicillin-binding proteins, participate in the synthesis and maintenance of cell membrane stability [19]. The SDS-PAGE electrophoresis results, depicted in Figure 6b, revealed notable differences in protein profiles between the MIC group and the control group. In the control group, the protein bands of *E. faecalis* appeared distinct and abundant. Conversely, the protein bands in the MIC group exhibited fainter colours, with some bands disappearing altogether. The macrolactin XY-treated group showed predominantly lighter protein bands, indicating that macrolactin XY (**1**) interfered with protein synthesis. This resulted in a decrease in the overall protein content of the bacteria, particularly affecting protein bands above 55 kDa, which nearly vanished. These observations suggest that macrolactin XY (**1**) primarily impacted the synthesis of larger-molecular-weight proteins.

#### 2.3.3. Effects on Cell Membrane Integrity

Cell membrane integrity is indicated by the leakage of nucleic acids from bacterial cells into the suspension [20]. As shown in Figure 6c, the nucleic acid content (measured as OD_260nm_) remained relatively stable and did not exhibit significant changes (*p* > 0.05) over time in both blank group and the negative control groups. However, a significant increase (*p* < 0.05) in nucleic acid content (OD_260nm_) was observed in the macrolactin XY-treated group, peaking at 0.563 in the 16th hour and remaining relatively unchanged.

#### 2.3.4. The Impact on the Expression of Genes Involved in Bacterial Energy Metabolism

Pyruvate kinase (PK) plays an important role in bacterial energy metabolism, serving as a crucial enzyme in the glycolytic pathway. It can catalyse the conversion of phosphoenolpyruvate to pyruvate, the process of which generates ATP, providing the necessary energy for bacteria. Furthermore, PK is involved in regulating the balance between glycogenolysis and gluconeogenesis to ensure that bacteria can flexibly adapt to changes in energy demand and maintain their normal life activities and physiological functions.

As shown in Figure 6d, it became evident that the expression of PK in *E. faecalis* was significantly downregulated (*p* < 0.0001) in the macrolactin XY-treated group at the MIC compared to the control group. This reveals that macrolactin XY modulates the transcript levels of genes encoding enzymes and proteins that play crucial roles in ATP synthesis, glycolysis, and other energy-producing processes.

## 3. Materials and Methods

### 3.1. General Experimental Procedures

NMR spectra were acquired using a Bruker AMX-500 spectrometer (Bruker Biospin Corp., Billerica, MA, USA), operating at 500 MHz for ^1^H-NMR and 125 MHz for ^13^C-NMR to ensure high-resolution and accurate spectral data. For HRESIMS analysis, an Agilent 6210 LC/MSD TOF mass spectrometer (Agilent Technologies Inc., Lake Forest, CA, USA) was employed, providing precise mass measurements. UV spectra were recorded on a UV-8000 spectrophotometer (Shanghai Metash instruments Co., Shanghai, China), ensuring the accurate quantification of absorbance values. Optical rotations were measured precisely using an Anton Paar MCP 5500 instrument (Anton Paar Co., Graz, Austria). Chromatographic separations were carried out on a Waters 1525 HPLC system (Waters Corp., Milford, MA, USA) equipped with a Waters 2996 photodiode array detector, utilizing YMC-Pack Pro C18 RS columns (5 µm) from YMC Co., Ltd. (Kyoto, Japan). Purifications were performed using an Agilent LC1263 Infinity II purification system (Agilent Technologies Inc., CA, USA). Thin-layer chromatography analysis was conducted on HSGF254-precoated silica gel plates (10–40 mm) from Yantai Chemical Plant (Yantai, China), ensuring the high-quality separation of analytes.

### 3.2. Bacterial Strain

The strain *Bacillus subtilis* sp. 18 was isolated from a co-epidermal sample collected from a sponge dwelling in the waters of the South China Sea at a depth of 325 m. This strain was being maintained by the Department of Biochemistry and Molecular Biology at the Naval Military Medical University. To propagate the strain, individual colonies were inoculated into test tubes containing 10 mL of ISP_2_ liquid medium (10 g/L of malt extract, 4 g/L of yeast, and 4 g/L of glucose). These cultures were then incubated for three days at a constant temperature of 28 °C with shaking at 180 rpm to generate seed cultures (OSMAC strategy in previous studies).

The Department of Biochemistry and Molecular Biology at the Naval Military Medical University generously provided us with a range of bacterial strains, including *E. faecalis* (ATCC 29212), *S. aureus* (ATCC 27217), *B. subtilis* (ATCC 21951), *E. coli* (ATCC 25922), *V. vulnificus* (ATCC 27562), and *V. parahaemolyticus* (ATCC 17802). To prepare these strains for subsequent experiments, 100 μL of each test strain was inoculated into test tubes containing 5 mL of nutrient broth medium (18 g/L of nutrient broth medium). The cultures were then incubated at 28 °C with shaking at 180 rotations per minute for 12 h, ensuring optimal activation and proliferation before proceeding to the next step.

### 3.3. Fermentation

The seed culture was added to 500 mL of ISP_2_ medium at a ratio of 10% (*v*/*v*) and incubated for eight days at 28 °C, with shaking at 180 rpm to allow for optimal growth and metabolism.

### 3.4. Extraction and Isolation

The fermentation broth underwent a separation process where it was filtered through eight layers of gauze, resulting in two distinct fractions. The filtrate was thoroughly mixed with an equal volume of ethyl acetate, while the bacterial material was thoroughly macerated with a combination of equal volumes of petroleum ether and ethyl acetate. Following stratification, the organic phase layer was collected, and this entire procedure was repeated three times. The combined organic phase layers from both the filtrate and the bacterial fraction were then evaporated using a rotary evaporator to obtain the fermentation product.

The crude extract (14 g) underwent vacuum liquid chromatography on a silica gel column, utilizing a gradient elution method with a mixed-solvent system of petroleum ether and ethyl acetate (from a 10:1 to 0:1) to obtain 10 fractions (A–Z). The fractions C, F, J, O, and Q underwent further elution on a forward silica gel column separately, employing a gradient of the same mixed-solvent system (from 50:1 to 0:1), yielding 13 subfractions for C (C1–C13), 10 subfractions for F (F1–F10), 4 subfractions for J (J1–J4), 12 subfractions for O (O1–O12), and 12 subfractions for Q (Q1–Q12). The fractions C12, C13, F2, F4, J2, O7, O9, Q7, and Q8 were then purified using HPLC (MeOH/H_2_O, 2.0 mL/min) separately, resulting in the purification of compound **2** (1.3 mg in C12)**,** compound **10** (1.6 mg in C13)**,** compound **11** (2.1 mg in F2), compound **9** (1.3 mg in F4), compound **8** (1.3 mg in J2), compound **5** (1.3 mg in O7), compound **6** (1.1 mg in O9), compound **1** (1.1 mg in Q7), compound **3** (2.0 mg in Q7), compound **4** (1.1 mg in Q8), and compound **7** (1.2 mg in Q8). The detailed isolation process for the known compounds is omitted for brevity.

Macrolactin XY (**1**): white powder; [*α*${]}_{D}^{25}$-93.90 (*c* 0.1, MeOH); UV (MeOH) *λmax* (log *ε*) 228 (3.68), 262 (3.36) nm; CD (MeOH) (∆*ε*) 233 (+13.16), 260 (−6.40); ^1^H-NMR and ^13^C-NMR data, see Table 1; HRESIMS *m*/*z* 439.2458 [M+Na]^+^ (calcd for C_25_H_36_O_5_Na, 439.2455) (Appendix A).

(5*R*, 9*S*, 10*S*)-5-(hydroxymethyl)-1,3,7-decatriene-9,10-diol (**2**): yellow oil; [*α*${]}_{D}^{25}$-15.2785 (*c* 1.0, MeOH); UV (MeOH) *λmax* (log *ε*) 226 (3.89) nm; CD (MeOH) (∆*ε*) 204 (+10.46), 224 (−10.03); ^1^H-NMR and ^13^C-NMR data, see Table 1; HRESIMS *m*/*z* 221.1150 [M+Na]^+^ (calcd for C_11_H_18_O_3_Na, 221.11482) (Appendix A).

### 3.5. ECD Calculations

Initially, the conformational analyses for compounds **1** and **2** were conducted utilizing the Spartan’10 software (Wave-function, Inc., Irvine, CA, USA) within the MMFF94 force field. Later, conformers with a Boltzmann population of over 5% were refined at the B3LYP/6-31+G (d, p) level, employing the conductor-like polarizable continuum model (CPCM) in MeOH. The theoretical ECD calculations for compounds **1** and **2** were calculated using the time-dependent density functional theory (TDDFT) approach at the B3LYP/6-311++G (2d, 2p) level in MeOH, respectively. The ECD spectra were generated by SpecDis 1.6 (University of Würzburg, Germany) using a Gaussian band shape with an exponential half-width of 0.3 eV, derived from dipole-length dipolar and rotational strengths.

### 3.6. Antibacterial Activity

#### 3.6.1. Determination of MIC and MBC in Macrolactin XY against Test Microorganisms

*E. faecalis* was activated according to the operation in 3.2 (to an optical density at OD_600_ of 0.6–0.8) and diluted with sterilized medium at a ratio of 1:1000. The sample was dissolved in methanol and further diluted with sterile water to achieve a concentration of 64 μg/mL. This experiment was carried out using the micro-broth dilution method; first, a sterile 96-well plate was prepared by filling the outermost circle with 200 μL sterile water to prevent the bacterial solution from drying, and then the bacterial solution was added to the remaining wells. An amount of 10 μL sample solution (at a concentration of 64 μg/mL) was added to the first well, and then diluted to 0.5 μg/mL by the multiplicative dilution method to make the final concentration in the range of 0.5–64 μg/mL, and the negative (methanol) and positive controls (levofloxacin) were set. The 96-well plate was incubated in a constant-temperature incubator maintained at 28 °C for 12 h.

#### 3.6.2. Analysis of Bacterial Growth Curve

*E. faecalis* was activated according to the operation in 3.2 (to an optical density at OD_600_ of 0.6–0.8) and the sample was dissolved in methanol and subsequently diluted with water to reach the MIC. To assess the growth curve, a sterile 96-well plate was prepared by leaving the outermost ring empty and filling each of the wells with 200 μL sterile water. In the remaining wells, 200 μL bacterial solution was dispensed, with 12 wells designated for each of the control and experimental groups. Then, 10 μL sample solution was received at the MIC of the experimental wells. The 9 plates were then incubated at 28 °C with agitation at 180 rpm. The optical density at 600 nm (OD_600_) of the wells was measured every two hours. This process was repeated three times, and the average value was recorded. Subsequently, the specific growth rate was computed for the logarithmic growth period using the formula ln(N2/N1) = μt (N2 and N1 represent the bacterial counts at different time points and μ is the specific growth rate).

### 3.7. Antibacterial Mechanism

#### 3.7.1. Bacterial Cell Membrane Potential

*E. faecalis* was activated according to the operation in 3.2 (to an optical density at OD_600_ of 0.6–0.8), and then transferred to a 1.5 mL EP tube for the next step. The MIC of compound **1** was added into the bacterial suspension, serving as the experimental treatment. Concurrently, a suspension lacking the sample solution was utilized as the blank control, while the solvent utilized to dissolve the sample represented the negative control. Following an incubation period of 10 h, the bacterial suspension was centrifuged at 4000 rpm for 10 min, yielding a bacterial pellet, which was washed twice with phosphate-buffered saline (PBS, 0.1 M, pH 7.2). Subsequently, a rhodamine 123 dye solution (rhodamine 123 powder was dissolved in methanol to form a solution of 1.0 g/mL and then diluted to 2.0 mg/mL with PBS) was added into the pellet, and then the mixture was incubated for 30 min in a dark area. All of the mixture was transferred to a 96-well plate, and the fluorescence intensity was measured using a fluorescence spectrophotometer (SpectraMax M2e, Molecular Devices, Sunnyvale, CA, USA).

#### 3.7.2. SDS-PAGE Electrophoresis of Bacterial Proteins

*E. faecalis* was activated according to the operation in 3.2 (to an optical density at OD_600_ of 0.6–0.8), and the bacterial solution was divided into two groups. One group (1 mL) served as the untreated control, while the other (1 mL) received 200 μL compound **1** at its MIC. Both groups were incubated for 10 h. After the incubation was completed, centrifugation was performed to remove most of the supernatant, and then an ultrasonic crusher (80% of maximum power, running for 15 s at 10 s intervals for 15 min) was used to crush the cells and release the proteins. The SDS-PAGE Gel Rapid Preparation Kit (Shanghai Yase Biotechnology Co., Ltd., Shanghai, China) was used to prepare the gel (loaded 10 μL protein sample). After electrophoresis was completed, Coomassie brilliant blue was applied to visualize the separated protein bands obtained after decolorization.

#### 3.7.3. Effects on Cell Membrane Integrity

The disruption of cell membrane integrity was demonstrated by quantifying the concentration of nucleic acids present in the bacterial suspension. *E. faecalis* was activated according to the operation in 3.2 (to an optical density at OD_600_ of 0.6–0.8), then dispensed into a 96-well plate (200 μL in each well), setting the blank control and negative control. For the experimental group, 20 μL sample solution containing an MIC was added, while the control group remained untreated. The incubation was carried out in an incubator at 28 °C, and 200 μL of the bacterial suspension was removed every 4 h. The OD_260_ value was determined using a UV–Vis spectrophotometer to indicate the nucleic acid content in the bacterial suspension.

#### 3.7.4. The Impact on the Expression of Genes Involved in Bacterial Energy Metabolism

Pyruvate kinase (PK), a crucial enzyme involved in bacterial energy metabolism, was chosen for further analysis. To design primers, the PK sequence of *E. faecalis* was retrieved through a database query. Additionally, the 16s rRNA gene was employed as an internal reference gene. *E. faecalis* was activated according to the operation in 3.2 (to an optical density at OD_600_ of 0.6–0.8). An amount of 2 mL bacterial suspension was removed and added to two EP tubes, respectively. For the experimental group, 200 μL of sample solution was added at an MIC, while the control group remained untreated and was incubated for 10 h for the next step. RNA extraction was carried out using the TransStart RNA Extraction Kit (Beijing Quan’s Gold Biological Company, Beijing, China), and reverse transcription was performed using the TransStart Top Green qPCR SuperMix Kit (Beijing Quan’s Gold Biological Company). The resulting cDNA was stored at −80 °C. Real-time fluorescence quantitative PCR (QuantStudio 5, Thermo Fisher Scientific, Waltham, MA, USA) was utilized for the analysis, and the CT values obtained were employed to assess the expression of pertinent genes using the 2-∆CT method. To guarantee the precision of the data, a minimum of three independent sets of experiments were conducted for each group.

### 3.8. Statistical Analysis

To ensure the accuracy and stability of the experimental data, all experiments were conducted in parallel groups and replicated three times. IBM SPSS Statistics 22.0 software (SPSS Inc., Armonk, NY, USA) was utilized for the analysis of the experimental data. One-way analysis of variance (ANOVA) was employed to determine significance levels, with *p*-values less than 0.05 considered statistically significant (* *p* < 0.05, ** *p* < 0.01, *** *p* < 0.001, **** *p* < 0.0001). To present the data in a more intuitive manner, GraphPad Prism was used for the statistical analysis of the experimental data.

## 4. Conclusions

In conclusion, eleven compounds, including two new compounds, were isolated from a fermentation broth of *Bacillus subtilis* sp. 18 through the OSMAC strategy in this study. Notably, a novel compound, macrolactin XY (**1**), demonstrated potent antibacterial activity against a diverse range of bacteria, particularly against *E. faecalis*, and its antibacterial mechanism was further investigated. Our findings suggest that the antibacterial mechanism of macrolactin XY (**1**) involves the disruption of the bacterial cell membrane, leading to the leakage of biomolecules such as nucleic acids and proteins, which disrupts the metabolic activity of the bacteria, ultimately leading to cell death (Figure 7). Furthermore, the mechanism also inhibited the expression of PK genes associated with bacterial energy metabolism, which disrupted the bacterial energy production pathway. Notably, a reduction in energy availability might adversely impact the functionality of ATP-reliant transporter proteins embedded in the bacterial cell membrane, particularly ion pumps and active transporters [21]. This, in turn, could disrupt the bacterial cell’s material transport mechanisms and osmotic pressure regulation, potentially affecting its overall metabolic health and physiological state (Figure 7). Regrettably, owing to the limited quantity of our compounds, only a single pivotal gene pertaining to energy metabolism was analysed. This work has laid a solid foundation for future studies on the antibacterial mechanism of macrolactin XY (**1**) and offers valuable insights for the development of macrolactins as potential antibacterial agents.

## Figures and Tables

**Figure 1 marinedrugs-22-00331-f001:**
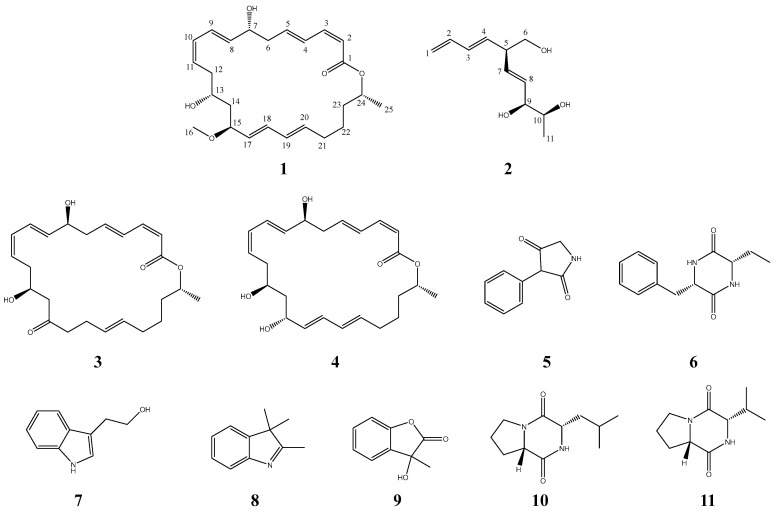
Structures of compounds **1**–**11**.

**Figure 2 marinedrugs-22-00331-f002:**
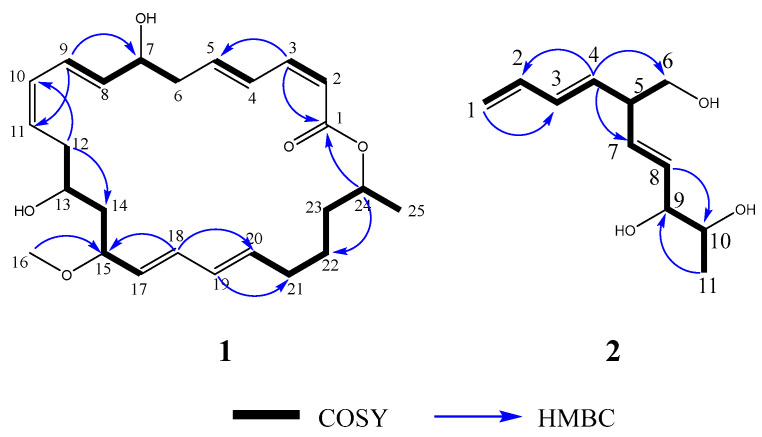
COSY and key HMBC correlations of compounds **1** and **2**.

**Figure 3 marinedrugs-22-00331-f003:**
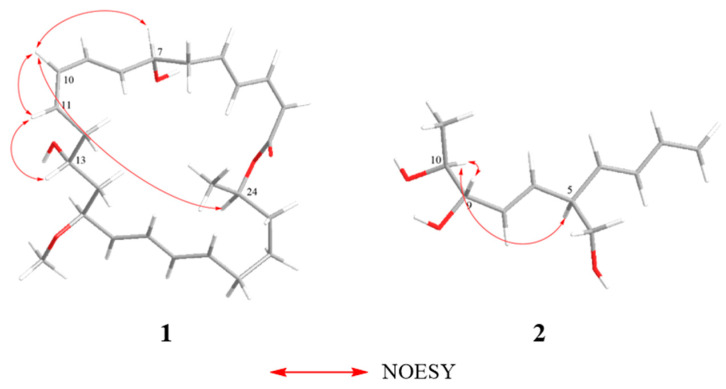
Key NOESY correlations of compounds **1** and **2**.

**Figure 4 marinedrugs-22-00331-f004:**
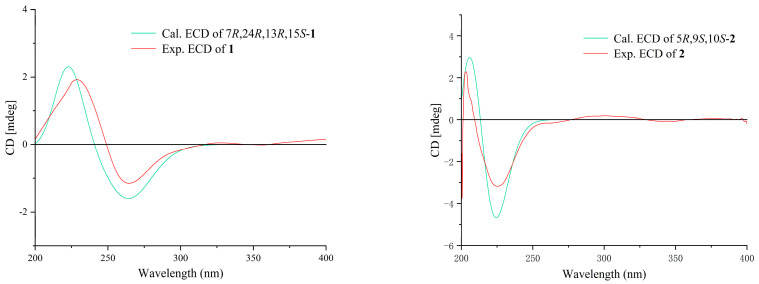
Calculated and experimental ECD spectra of compounds **1** and **2**.

**Figure 5 marinedrugs-22-00331-f005:**
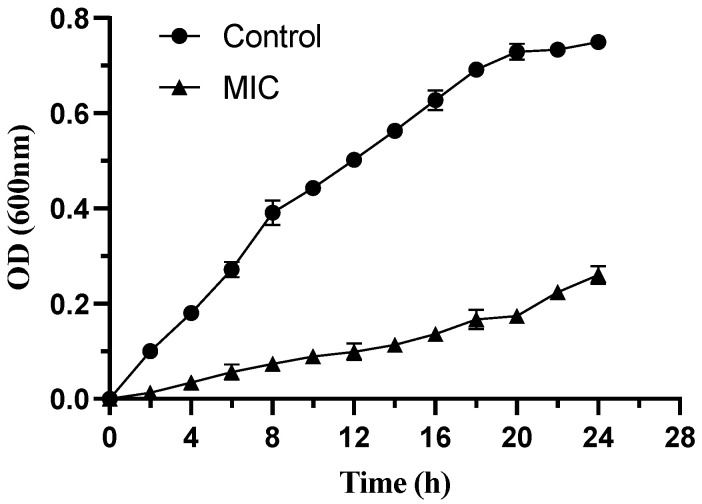
Growth curve of *E. faecalis*.

**Figure 6 marinedrugs-22-00331-f006:**
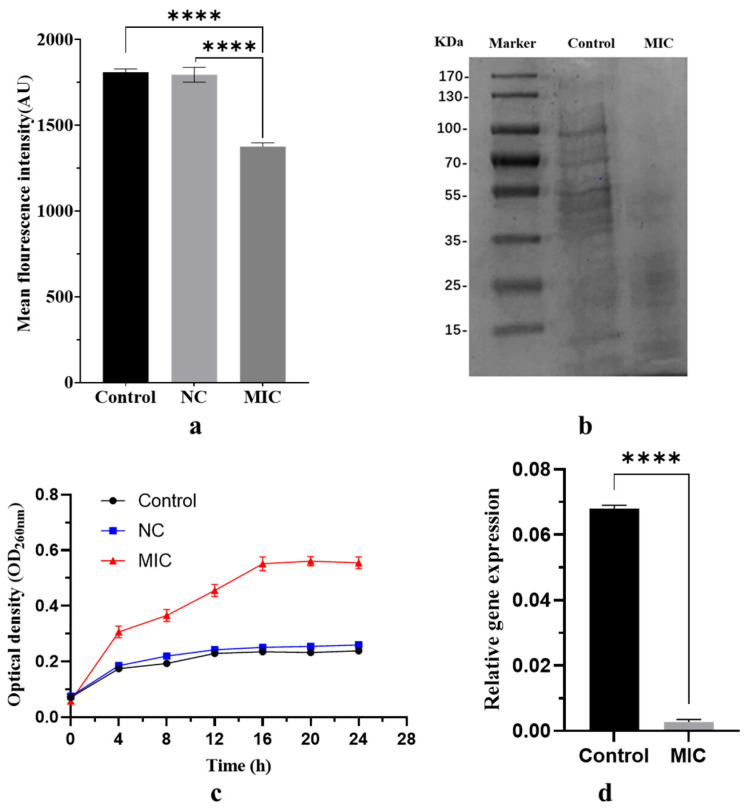
(**a**) Changes in fluorescence intensity of *E. faecalis* after addition of macrolactin XY. “****” represents *p* < 0.0001. (**b**) The gel electrophoresis picture of *E. faecalis* intracellular proteins. (**c**) Effect of macrolactin XY on cell membrane integrity of *E. faecalis*. (**d**) Relative expression of PK in *E. faecalis* following the action of macrolactin XY.

**Figure 7 marinedrugs-22-00331-f007:**
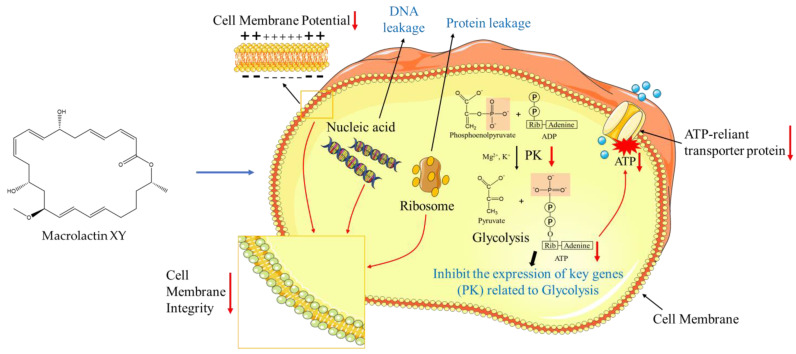
Possible antibacterial mechanism of action of macrolactin XY.

**Table 1 marinedrugs-22-00331-t001:** ^1^H and ^13^C-NMR data of compounds **1** (in CDCl_3_) and **2** (in CD_3_OD).

Position	1 *		2 *	
	*δ*c, Type	*δ*_H_, Mult. (*J* in Hz)	*δ*c, Type	*δ*_H_, Mult. (*J* in Hz)
1a	166.4, C		116.2, CH_2_	4.99, dd, (10.1, 1.8)
1b				5.13, dd, (14.3, 1.5)
2	118.2, CH	5.59, d, (11.6)	138.5, CH	6.35, dt, (14.2, 8.5)
3	143.0, CH	6.54, m	133.6, CH	6.15, dd, (13.0, 8.5)
4	130.0, CH	7.24, dd, (11.4, 15.2)	135.1, CH	5.67, m
5	139.6, CH	6.07, m	49.7, CH	2.98, t, (6.9)
6	41.7, CH_2_	2.47, m	66.3, CH_2_	3.53, d, (6.6)
7	71.4, CH	4.35, q, (6.9)	133.5, CH	5.62, m
8	135.8, CH	5.75, dd, (5.5, 15.1)	132.5, CH	5.58, dd, (5.5, 13.0)
9	125.1, CH	6.58, m	77.9, CH	3.86, m
10	130.0, CH	6.09, m	71.5, CH	3.65, m
11	128.3, CH	5.53, dt, (9.0, 9.0)	18.7, CH_3_	1.14, d, (6.4)
12	35.4, CH_2_	2.42, m		
13	69.4, CH	3.91, m		
14a	40.4, CH_2_	1.77, m		
14b		1.71, m		
15	80.4, CH	3.96, m		
16	56.4, CH_3_	3.27, s		
17	130.1, CH	5.43, dd, (8.0, 15.2)		
18	133.3, CH	6.15, dd, (10.4, 15.2)		
19	129.9, CH	6.02, m		
20	135.5, CH	5.68, dt, (7.0, 14.5)		
21a	32.1, CH_2_	2.19, m		
21b		2.09, m		
22	24.6, CH_2_	1.51, m		
23	35.0, CH_2_	1.64, m		
24	71.2, CH	5.03, m		
25	20.0, CH_3_	1.28, d, (6.3)		

***** 500 MHz for ^1^H-NMR and 125 MHz for ^13^C-NMR.

**Table 2 marinedrugs-22-00331-t002:** Antibacterial activity of compounds **1**–**11** against various indicator bacteria.

Compound	MIC (μg/mL)					
	*E. coli*	*E. faecalis*	*B. subtilis*	*S. aureus*	*V. traumaticus*	*V. parahaemolyticus*
**1** *	6	3	6	6	-	-
**2** *	6	-	-	6	-	-
**3**	-	-	6	6	-	-
**4**	-	-	6	6	-	-
**5**	-	-	-	12	-	-
**6**	6	-	-	-	-	-
**7**	-	-	-	-	-	-
**8**	-	-	12	-	-	-
**9**	12	-	-	6	-	-
**10**	12	-	-	-	-	-
**11**	-	-	-	-	-	-
Levofloxacin	0.5	0.5	0.5	0.5	0.5	0.5
Methanol	-	-	-	-	-	-

“*” indicates new compound; “-” indicates inactive.

**Table 3 marinedrugs-22-00331-t003:** Specific growth rate of *E. faecalis* in control and administered groups.

Time	Control Group	MIC Group
2 h	1.29 ± 0.02	0.31 ± 0.02
4 h	0.77 ± 0.03	0.39 ± 0.03
6 h	0.61 ± 0.01	0.35 ± 0.01
8 h	0.51 ± 0.02	0.29 ± 0.01
10 h	0.42 ± 0.03	0.25 ± 0.02
12 h	0.36 ± 0.02	0.22 ± 0.03
14 h	0.31 ± 0.03	0.19 ± 0.02
16 h	0.28 ± 0.01	0.18 ± 0.01
18 h	0.26 ± 0.02	0.17 ± 0.02
20 h	0.23 ± 0.02	0.16 ± 0.01
22 h	0.21 ± 0.01	0.15 ± 0.02
24 h	0.19 ± 0.01	0.14 ± 0.03

## Data Availability

The data presented in this study are available on request from the corresponding author.

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
