# Peer review of "Macrolactin X, a Macrolactin Antibiotic from Marine Bacillus subtilis sp.18"

_marinedrugs, 2024, doi:10.3390/md22080331_

Round 1

Reviewer 1 Report

Comments and Suggestions for Authors

Macrolactins showed antibacterial activity against a panel of pathogens Staphylococcus aureus, Bacillus subtilis, and Escherichia coli. This manuscript described the isolation, structure elucidation, antibacterial activity, and primary mechanism of new analogue from marine-derived Bacillus subtilis sp. 18. The story is interesting. Some issues should be addressed before it be processed further.

Title: Novel often refers to an unpublished carbon skeleton, please delete it.

Abstract:

1.       Compound 1-5 >> Compounds

2.       destroy the disruption: There is some redundance.

3.       It may destroy the disruption of 15 bacterial cell membrane integrity and permeability, and inhibit the expression of genes associated 16 with bacterial energy metabolism by cell membrane potential, SDS-PAGE electrophoresis, cell mem-17 brane integrity and key genes expression experiments.

Please rewrite this sentence. It may be method ‘by cell membrane potential, SDS-PAGE electrophoresis, cell mem-17 brane integrity and key genes expression experiments’. The description isn’t clear.

Main text:

4.       Figure 1: It is not necessary to indicated the new compounds.

5.       Page 2, line 59: Normally the compound name was given at the end of the structure elucidation.

6.       Page 2, line 63: 1.29 (3H, m). It is doublet with J of 6.32 Hz in table 1. ‘one characteristic methyl proton signals’, but there are two signals after these words.

7.       Page 2, line 72-73: C-1/C-22/C-23, H-3 to C-1/C-5, H-9 to C-7/C-10, 72 H-16 to C-15, H-12 to C-10/C-11/C-13/C-14, and H-18 to C-17/C-19/C-21. ‘/’ normally refers to adjacent carbons, so some of them should be changed to ‘and’. And check all the text.

8.       Page 3, line 79: substituention. Substitution. It is hydroxy here in reported compound. If so, try to compare the NMR data, optical rotation and CD to determine the configurations.

9.       Page 4, line 84-88: The NOESY description should be rewrote. For the macrocycle compound, the NOESY signals sometimes aren’t correct.

10.    Page 5, line 120-121: Please carefully check the NOESY in a structure with free rotation bonds. For example, carbon bond C-5/C-6 can free rotation, so the NOESY signal between H-5 and H-6 may come from COSY.

11.    Page 5, line 129: Compound 1-11. Compounds. Please check all the plural form.

12.    Page 5, line 132-133: macrolactin F (2), and macrolactin A (3). They are compounds 3 and 4?

13.    Page 6, line 148: E. faecalis. Please use Italic.

14.    Page 6, table 3: The digits after the decimal point should be same.

15.    Please incorporate the experimental data regarding the minimum bactericidal concentration (MBC) of compound 1 into the section 2.2.1 “Determination of MIC and MBC in macrolactin X against Test Microorganisms”.

16.    During experiments investigating the impact of compound 1 on gene expression, the focus was solely on pyruvate kinase, while other crucial genes, including citrate synthase and isocitrate dehydrogenase should be performed.

17.    The picture layout could be improved for optimal clarity. It is advisable to merge figures 5-9 to consolidate the antibacterial mechanism. Furthermore, in figure 7, the labels "marker," "control," and "MIC" should be displayed at the top of the figure.

18.    Why does the leakage of nucleic acids serve as an indicator of membrane condition? Furthermore, is there any consideration of the potential influence of nucleases and similar enzymes on the experimental outcomes?

Comments on the Quality of English Language

Minor English language should be made. 

Author Response

Thank you very much for taking the time to review this manuscript, your comments have greatly benefited our manuscript. We have carefully checked this paper according to all constructive suggestions and answered all the questions raised in the comments.

Reviewer 2 Report

Comments and Suggestions for Authors

This manuscript reported a detailed investigation on the marine bacterial Bacillus subtilis sp.18, resulting in the isolation of two new compounds macrolactin X (1) and dihydroxylaureonitol (2), along with nine known analogues. Their structures were established by the extensive analysis of NMR spectral data. Furthermore, the absolute configurations of two new compounds were determined by ECD calculations. Interestingly, macrolactin X (1) displayed significant antibacterial activity against Enterococcus faecalis, and the mechanism was extensively investigated. In summary, these findings were important, and this manuscript was well-organized, which worth to be published in this journal.

However, revisions were required as followings:

1. For compound 1:

1.1 P2L62: There were two methyls, not ‘one characteristic methyl proton signals’.

1.2 P2L63: The splitting pattern in ‘δH 1.29 (3H, m)’ was not consistent with that recorded in Table 1.

1.3 As shown in Figure S5, it seemed that the COSY correlation H-4/H-5 could be observed. Please check it.

1.4 It was important to provide the HMBC correlation from H-25 to C-24 in the main text, which could be clearly observed in Figure S7, to support ‘the carbon at C-24 was substituted with methyl’.

1.5 The configurations of double bonds should be determined in the main text.

2. For compound 2:

2.1 P2L62: There were seven olefinic protons, not ‘six double-bonded proton signals’.

2.2 The expression ‘methylene oxide’ was not appropriate, perhaps ‘oxygen-bearing methylene’ was better.

2.3 As shown in Figure S15, the COSY correlation H-4/H-5/H-6 could be observed. Please check it.

2.4 The configurations of double bonds should be clearly in the main text before calculations.

3. Due to the flexibility of macro-ring and the linear chain, it is risky to assign the relative configurations of compounds 1 and 2 depend on the NOESY correlations. For compound 1, perhaps hydrolysis of the methyl ether to yield 1,3-diol. Then this diol reacted with acetone to give a six-member ketal product, which would be benefit to determine the relative configuration on the basis of NOESY correlations. For compound 2, the similar ketal product could be obtained. And did you try to use the Mosher’s method to determine the absolute configurations of C-7 and C-13 in compound 1 and absolute configurations of C-5, C-9 and C-10 in compound 2?

4. What were the names for the known compounds 311? And check whether ‘macrolactin F (2) and macrolactin A (3)’ were correct or not?

5. The detailed isolation process for the new compound 2 should be provided. And it is better to provide the detailed isolation process for the known compounds.

6. In the experiment, the UV data for compounds 1 and 2 were interchanged, according to Figures S1 and S11.

7. ‘rhodamine powder was dissolved in methanol to form a solution of 1.0 mg/mL and then diluted to 2 g/mL with PBS’. It seemed it was not possible to dilute 1.0 mg/mL to 2 g/mL. Please check it.

Others:

1. P1L12: Compound 1-5 and 7-10Compounds 15 and 710

2. P1L13&P5L132: ranging from 3-12 μg/mLranging from 3 to 12 μg/mL

3. P2L64: methynesmethines

4. P3L79: substituentionsubstitution

5. Table 1 caption: compounds 12compounds 1 and 2

6. P4L93: Italic font missing ‘7R, 13R, 15S, 24R’ ‘7R,13R,15S,24R

7. P4L94: Bold font missing ‘ECD spectrum of 1’ ‘ECD spectrum of 1

8. Figure 4 legend: ‘7R, 24R,113R,5S-1 ‘7R,13R,15S,24R-1

caption: compound 12compounds 1 and 2

9. P5L110: potential e of producingpotential of producing

10. P5L129: Compound 1-11Compounds 111

11. P5L131: ‘Compound 1-5 and 7-10compounds 15 and 710

12. Table 2 caption: compound 1-11compounds 111

13. P6L138: macrolactin A and Fmacrolactins A and F

14. P6L148: Italic font missing E. faecalisE. faecalis

15. P11L280&L284: compound 1 and 2compounds 1 and 2

16. P11L294: 64 μg/ml64 μg/mL

Comments on the Quality of English Language

Typo and grammar errors were observed, some of which were given in the comments.

Author Response

(The authors gave the same response as above.)

Round 2

Reviewer 1 Report

Comments and Suggestions for Authors

1. Compound 2,H-2: Although the 1H NMR is smearing, it looks like ddd with coupling constants of 17.0,10.3,10.3 Hz. Please double check and confirm the geometry configuration of the double bonds.

2.  Please carefully check the NORESY signals for the free rotation moiety.

3.   Please check the NOESY signal for H21a-H15 in SI. Also check if the other NOESY signals description in main text present in NOESY spectrum.

Author Response

(The authors gave the same response as above.)
